# Characterization of Electrospinning Prepared Nitrocellulose (NC)-Ammonium Dinitramide (ADN)-Based Composite Fibers

**DOI:** 10.3390/nano13040717

**Published:** 2023-02-13

**Authors:** Qiong Wang, Lu-ping Xu, Chong-qing Deng, Er-gang Yao, Hai Chang, Wei-qiang Pang

**Affiliations:** Xi’an Modern Chemistry Research Institute, Xi’an 710065, China

**Keywords:** nitrocellulose, energetic material, electrospinning, ammonium dinitramide

## Abstract

Nanoscale composite energetic materials (CEMs) based on oxidizer and fuel have potential advantages in energy adjustment and regulation through oxygen balance (OB) change. The micro- and nanosized fibers based on nano nitrocellulose (NC)-ammonium dinitramide (ADN) were prepared by the electrospinning technique, and the morphology, thermal stability, combustion behaviors, and mechanical sensitivity of the fibers were characterized by means of scanning electron microscope (SEM), transmission electron microscopy (TEM), differential scanning calorimetry (DSC), gas pressure measurement of thermostatic decomposition, laser ignition, and sensitivity tests. The results showed that the prepared fibers with fluffy 3D macrostructure were constructed by the overlap of micro/nanofibers with the energetic particles embedded in the NC matrix. The first exothermic peak temperature (T_p_) of the samples containing ADN decreased by 10.1 °C at most compared to that of ADN, and the pressure rise time of all the samples containing ADN moved forward compared to that of the sample containing NC only. Furthermore, ADN can decrease the ignition delay time of NC-based fibers under atmosphere at room temperature from 33 ms to 9 ms and can enhance the burning intensity of NC-based fibers under normal pressure. In addition, compared to the single high explosive CL-20 or RDX, the mechanical sensitivities of the composite materials containing high explosive CL-20 or RDX were much decreased. The positive oxygen balance of ADN and the intensive interactions between ADN and NC can reduce the ignition delay time and promote the burning reaction intensity of NC-based composite fibers, while the mechanical sensitivities of composite fibers could be improved.

## 1. Introduction

Nitrocellulose (NC), fabricated by the nitrification of cellulose, has been widely used in military and commercial fields [1,2], such as for gun propellants [3], solid rocket propellants [4,5], explosives [6,7], lacquer [8], and coatings [9,10,11]. The reason for its extensive application in the military field is that NC possesses higher formation enthalpy (−2767.29 J/g for NC with nitrogen content 11.97%) than that of other polymers, such as GAP (142.8 J/g), BAMO (428.4 J/g) and AMMO (99.6 J/g) [12]. In addition to possessing high energy, having the controllable reaction rate and acceptable safety are another two requirements for energetic materials used in military ammunition [13]. The preparation of nanosized energetic materials has made it possible to achieve those targets, owing to the bigger surface to volume ratio and the higher reactivity that the nano-energetic materials have compared to common micro-energetic materials [14,15,16]. The higher reactivity of nanosized NC has great potential to improve the combustion performance of propellants or explosives [17,18]. In addition to that, as common NC fibers are downscaled to nanosized particles, the friction sensitivity can be improved substantially, with the simultaneous increase in the burning rate [19]. In particular, the nano-crystallization of spherical NC particles with diameter = 15–30 nm using the composite crystal method can significantly decrease the impact sensitivity by 44.6% [20]. In many methods of nanosized energetic materials preparation, electrospinning is a selective preference due to its versatile choice of raw materials, the relatively economic hardware, the superior design flexibility of the composition, and the feasibility of mass production [21,22]. In recent years, many micro/nanosized NC-based composites have been designed and produced by electrospinning in order to improve the combustion performance [23,24], to reduce the aggregation degree of the nanoparticles [25,26], or to decrease the sensitivity of high energetic explosives, such as hexahydro-1,3,5-trinitro- 1,3,5-triazine (RDX) [27,28]; octahydro-1,3,5,7-tetranitro-1,3,5,7- tetrazocine (HMX) [29,30]; or 2,4,6,8,10,12- hexanitro-2,4,6,8,10,12- hexanitrohexaazaiso-wurzitane (CL-20) [31]. However, those electrospun composites reported above were composed of nitrocellulose and high explosives which had a negative oxygen balance (OB), such as RDX (OB = −21.6%), HMX (OB = −21.6%), CL-20 (OB = −10.9%), et al. It was reported that compounds with an OB close to zero demonstrate increased energy release [32]; hence, it is expected that the energy release rate can be improved by adding oxidizer into those electrospun composites. Furthermore, the addition of materials with high reactivity or high energy, such as thermite [33] or explosive [28], into the NC polymer matrix can result in a significant increase in combustion propagation velocity and the total energy release. The utilization of the high energetic oxidizer, such as ammonium dinitramide (ADN), gives several desirable characteristics when employed in solid rocket propellants and explosives. In this work, ADN [34,35], one kind of green oxidizer with high energy and an OB of 25.8%, was introduced into the NC matrix to compose a system with high reactivity that behaves in the way that thermite does. Additionally, the high explosives RDX and CL-20 were also introduced to improve the total energy. As far as we know, the preparation of the electrospun NC-based composites containing ADN together with high explosives and the influence of ADN on the performance of electrospun NC-based fibers has not been reported on yet.

NC/ADN-based composite fibers containing RDX and CL-20 were prepared by electrospinning. The morphology of the as-prepared fibers was characterized by scanning electron microscopy (SEM) and transmission electron microscopy (TEM), respectively. Furthermore, the thermal stabilities were carried out, respectively, by the differential scanning calorimetry (DSC) and the gas pressure measurement of thermostatic decomposition. In addition, the combustion performances and the mechanical sensitivities were also investigated.

## 2. Materials and Methods

### 2.1. Raw Materials

NC (N% = 11.97%), ADN, RDX, and CL-20 were provided by Xi’an Modern Chemistry Research Institute. The acetone: (AR grade, Si Chuan Xilong scientific Co., Ltd., ChengDou, SiChuan, China). The ethanol: (AR grade, Sinopharm Chemical Reagent Co., Ltd. ShangHai, China).

### 2.2. Preparation of the Spinning Solution

The NC solution with the concentration of 10 wt%, as the control sample, was prepared by dissolving certain dry NC fibers into acetone and oscillating for 40 h until it formed a transparent and homogenous solution. The RDX solution (40 wt%) and CL-20 solution (40 wt%) in acetone were prepared by dissolving the solutes in acetone, respectively. The precursor solutions for electrospinning were prepared as follows: 30 mg ADN were loaded in a 1.5 mL bottle, and 0.5 g NC solution (10 wt%) was added, then, the additives in the form of solutions were added in the as-prepared NC-ADN solution. All the precursor solutions were oscillated for 24 h before use. The components of each sample are listed in Table 1.

### 2.3. Electrospinning Process

The precursor solution was loaded into a plastic syringe with a needle inner diameter of 0.51 mm, and the syringe was fixed horizontally on a syringe pump. During the process of electrospinning, the high positive voltages 6 kV (for sample 1–3) and 12 kV (for sample 4) were imposed on the needle, respectively. Aluminum foil acted as the collector and was placed at about 5 cm away for sample 1–3 and 10 cm away for sample 4, under the needle and collected to the ground. The needle and the syringe were connected by Teflon tube with the outer diameter of 2.0 mm, and the flow rate was set at 2.5 mL/h.

### 2.4. Morphology Analysis

Filed emission scanning electron microscopy (SEM) with the model JSE-5800 and transmission electron microscopy (TEM) with the model FEI Talos F200X were used to observe the morphology of the electrospun NC-based fibers.

### 2.5. DSC Tests

The thermal decomposition of the samples was investigated by means of differential scanning calorimetry (DSC, Q200, TA instruments, Newcastle, USA), with the heating rate of 10 °C/min and the sample mass of 0.5 mg under a high-purity nitrogen atmosphere with the flow rate of 25 mL/min. All the tests were performed in curling alumina crucibles. In this section, the prepared samples and energetic materials, such as NC, ADN, RDX, CL-20, and the NC+ADN mixture (mass ratio 1/1), were made.

### 2.6. Gas Pressure Measurements of Thermostatic Decomposition

The gas pressure of thermal decomposition for each sample under thermostatic conditions was measured by a self-developed instrument with the diagrammatic sketch shown in Figure 1. The constant temperature of the furnace was kept to 100 ± 0.2 °C; the precision of the gas pressure measurement was about ±1%. About 16 mg of sample was put into a glass reactor which connected to the transducer through the adapter. Then, the as-prepared reactor was put into the heating hole in a thermostat. Simultaneously, the pressure was recorded by the data acquisition instrument as a pressure vs. time curve. The initial pressure was equal to the atmosphere. For comparison, the NC+ADN mixture (10 mg/6 mg, the same mass ratio as sample 3), prepared by simple mixing, was tested.

### 2.7. Combustion Tests

The combustion behaviors of the samples were characterized by a self-developed instrument with the schematic diagram shown in Figure 2; it was composed of a solid CO_2_ laser (SLC110, 10.6 μm), a photo detector, a high-speed camera, and a digital oscilloscope (TEK DPO4034). The ignition delay time was calculated by the difference between the moment the laser emitted and the moment that the light of the igniting sample reached the photo detector. Simultaneously, the combustion processes were recorded by the high-speed camera with 2000 fps. The sample with the dimensions of roughly 5 mm × 5 mm was pasted on the combustion cell, and each sample was tested three times with the average as the ignition delay time.

### 2.8. Mechanical Sensitivity

The impact sensitivity was measured on the WL-1 instrument with the hammer weight of 10 kg, the sample mass of 50 mg, and the drop height of 25 cm. Each sample was pressed into a pill by a tablet machine before the measurement. In total, 25 tests were conducted for each sample, and the explosion probability was calculated as the last result in the form of an explosion percentage. The friction sensitivity was measured on the WM-1 instrument with the pressure of 3.92 MPa, a swing angle of 90°, and the sample mass of 30 mg. Each sample was pressed into a pill by a tablet machine before the measurement. Similarly, 25 tests were conducted for each sample, and the explosion probability was calculated as the last result in the form of an explosion percentage.

## 3. Results and Discussions

### 3.1. Morphology Characteristics

The morphology of the electrospun NC-based composite fibers characterized by SEM and TEM is shown in Figure 3. The optical photographs of the electrospun NC-based composite fibers are shown in Figure 4.

It can be seen in Figure 3 that all the NC-based composites were composed of crossed, overlapped fibers in the order of hundreds of nanometers, microscopically. Many fiber bundles can be found in the SEM images of samples 1–3 containing ADN, However, single fibers with a smooth surface can be seen in sample 4 without ADN. As shown in the TEM images in Figure 3, solid particles were found to be embedded in the NC matrix. Macroscopically, as shown in Figure 4a–c, NC-based composites with a fluffy 3D macrostructure were fabricated by the addition of ADN and had cotton-like shapes. However, the electrospun NC without ADN is just a 2D web (Figure 4d). The vertical growth of fibers can be seen, and many tiny fibers were formed on the outside of the needle outlet (Figure 4e).

Traditionally, the electrical field force overcame the surface tension and dragged the drops forward; then, the drops flew to the collector. In this process, as the solvents evaporated quickly, the gradually dried fibers reached the collector and overlapped layer by layer, forming a non-woven fabric. Although the mechanism of promoting the formation of a 3D macrostructure by the addition of ADN is still under exploration, the following points can be ascertained [36]. Firstly, the surface tension would be much reduced compared to the spun solution without the ADN addition. Furthermore, the surface charge would be much increased compared to the spun solution without the ADN addition. Last but not least, the solution conductivity would be improved a lot. Those factors synergistically improved the repulsion force on the liquid surface, which caused the difficult formation of the Tyler cone and the stable jet flow like in traditional electrospinning. Phenomenally, as shown in Figure 4e, the jet containing ADN preferred to disperse into many tiny jets at the needle outlet, which accelerated the vaporization of the solvent. The electrostatic repulsion on the jet surface and the gradually dried fibers made the jets and the fibers scatter haphazardly, which resulted in the complete solvent vaporization a long time before the fiber reached the collector; finally, they contributed to the formation of fluffy 3D macrostructure.

### 3.2. Thermal Behaviors under Linear Heating Conditions

The thermal behaviors of samples 1–4 at linear heating rate of 10 °C/min were tested; the DSC curves are shown in Figure 5, and the peak temperature data of the main decomposition processes are listed in Table 2. 

It can be seen in Figure 5 that the melting point of ADN and RDX disappeared for the NC-based electrospun samples, compared to the DSC curves of ADN and RDX, which can be attributed to the ADN and RDX particles that were embedded in the fluffy NC nanometric fibers (Figure 4). Compared to the pure NC, the thermal decomposition peak temperature (T_p_) of the NC in sample 4 moved forward by 1.1 °C, much less than the decrement of electrospun NC fiber reported in [28], indicating that the thermal stability of the NC fibers obtained by traditional electrospinning can be retained. For samples 1–3 containing ADN, the T_p_ of the ADN decreased from 189.0 °C to 178.9 °C, 184.5 °C, or 185.7 °C, respectively, compared to the pure ADN. The T_p_ of NC decreased from 210.1 °C to 201.5 °C, 204.3 °C, or 202.1 °C, respectively, compared to the pure NC. The T_p_ decrease indicated that reactions happened between ADN and NC in heating. It can be seen that the T_p_ of NC+ADN decreased by 24.2 °C compared to the pure NC, and 3.1 °C compared to the pure ADN, which showed that the intensive interactions existed between ADN and NC. Comparing the DSC curves of sample 3 to that of the NC+ADN mixture, it can be seen that a unique T_p_ of 185.9 °C appeared in the DSC curves of the NC+ADN mixture; however, two T_p_s, corresponding to ADN (185.7 °C) and NC (202.1 °C) appeared in that of sample 3. The difference between these two samples with the same components may be caused by the different NC/ADN mass ratio in the samples. The same phenomenon can be observed for samples 1 and 2. With the addition of ADN, the exact intensive interactions between ADN and NC were recognized as the main reasons for the T_p_ decrease in ADN and NC in samples 1, 2, and 3. The T_p_ of the RDX in the NC-based fibers (sample 1) decreased by 14.2 °C compared to the pure RDX. This large T_p_ decrease in RDX in sample 1 can be ascribed to the following two aspects. The first one is that RDX was prone to decompose when the particle size was reduced from the micrometer to the nanometer scale [37]. The second one is the acceleration influence of ADN on the thermal decomposition of RDX. Similarly, the T_p_ of CL-20 in the NC-based fibers containing CL-20 (sample 2) decreased from 251.3 °C to 235.4 °C, which is about 4.2 °C lower than that reported by Guo [38].

### 3.3. Thermal Behaviors under Thermostatic Conditions

The thermal behaviors of samples 1–4 at 100 °C were tested by the gas pressure measurement in the process of thermal decomposition, and the pressure curves are shown in Figure 6.

It can be seen in Figure 6 that the pressure of sample 4 containing only NC did not change much in the whole heating process, which was much lower than that of samples 1, 2, and 3. Furthermore, the thermal decomposition can be accelerated after being heated for 1.5 h, 2.0 h, and 1.1 h for samples 1, 2, and 3, respectively, which were much shorter than the initial pressure rise time of sample 4 containing NC alone. The first pressure rise process for samples 1, 2 and 3 can be ascribed to the thermal decomposition of ADN because ADN decomposed ahead of NC, as shown in the DSC curves in Figure 5. Obviously, as ADN was introduced the thermal decomposition of the fibers was accelerated. This was due to the intense interactions between ADN and NC, as analyzed in the previous section. Comparing the pressure curve of sample 3 with that of sample 4, as shown in Figure 6, the intensive interactions between ADN and NC can also be ascertained. As the high-energy compounds, CL-20 or RDX, were additionally added, the pressure rise time was delayed a little. Apart from that, a comparative study on the decomposition behavior of the NC+ADN mixture with that of sample 3 was conducted. In Figure 6, it can be seen that the pressure of the NC+ADN mixture began to rise after heating for 30 h, which was about 28.9 h later than that of sample 3, which indicated that the more intensive interactions between ADN and NC can be prompted by electrospinning. This acceleration effect brought by electrospinning can be ascribed to better dispersion uniformity than simple mixing and the resulting large contact area between ADN and NC.

### 3.4. Combustion Performance

The ignition delay times of samples 1, 2, 3, and 4 were measured with the laser powers of 40 W and 67 W, respectively, and the results are shown in Figure 7.

It can be seen in Figure 7 that as the laser power increased, the ignition delay time of all the samples decreased, and the differences among those four samples decreased. Under a laser irradiation of 40 W, the order of the ignition delay time was [sample 8 (33 ms)] ≫ [sample 1 (14.5 ms)] > [sample 2 (9 ms)] = [sample7 (9 ms)]. However, as the power increased to 67 W, the order of the ignition delay time was [sample 8 (14.5 ms)] ≫ [sample 1 (7.5 ms)] ≈ [sample 2 (7 ms)] ≈ [sample7 (6 ms)]. The pure NC electrospun fibers had the longest ignition delay time under these two laser power irradiations. As ADN was added separately, the ignition delay time decreased the most. When CL-20 and ADN were added together to the NC-based matrix, the ignition delay time changed a little compared to that of the NC-based fibers containing and added alone, but decreased a lot compared to that of the pure NC electrospun fibers. The phenomenon was a little different when RDX and ADN were added together to the NC-based matrix. The ignition delay time changed a little under the laser power of 67 W and increased a lot under the laser power of 40 W compared to that of the NC-based fibers with ADN alone. Compared to the results reported by Wang [27] that the ignition delay time of the electrospun NC fibers containing RDX was higher than that of the NC fibers, it is easy to conclude that the addition of ADN can substantially reduce the ignition delay time of NC fibers containing RDX. Due to the laser ignition starting with the thermal decomposition of the material, the one with the lower thermal decomposition temperature was more inclined to ignite [39]. As shown in Figure 5, the first decomposition peak temperatures of the samples containing ADN (178.9 °C, 184.5 °C, and 185.7 °C for sample 1, 2, and 3, respectively) were much lower than that of sample 4 (209.0 °C); hence, the ignition delay time of samples 1–3 was shorter than that of sample 4. However, this was not true for the three samples 1–3 that contained ADN; for example, sample 1, with the lowest first decomposition temperature (178.9 °C), had the longest ignition delay time. The reason was that RDX had a retarding effect on the laser ignition process of the NC composite fibers [27], which was consistent with the reported results [40,41]. Secondly, the intensive interactions between ADN and NC made the ADN addition play the critical role in the igniting tests.

The combustion processes were captured with a high-speed camera during the laser ignition tests, and some of the corresponding snapshots are shown in Figure 8.

The combustion flames were weak in the whole process for sample 4 composed of pure NC, and the combustion flames became brighter as ADN, RDX, or CL-20 was added, as shown in Figure 8. This indicated that the burning became more violent as the composition became more energetic when high explosives were added into the NC matrix. This phenomenon was consistent with the ignition delay time results. The reason for the improvement in the burning intensity by adding ADN is that the energetic salt possesses positive oxygen balance [42]. From the ignition delay time and combustion flame in the burning process, it can be seen that the combustion performance of NC-based composite fibers can be improved significantly by the addition of ADN.

### 3.5. Mechanical Sensitivity

The impact sensitivity and friction sensitivity of samples 1, 2, 3, and 4 were tested and compared with those of pure ADN, RDX, and CL-20. The results are listed in Table 3.

It can be seen that the impact sensitivity and friction sensitivity for samples 1 and 2 were much reduced compared to those of pure RDX and CL-20, which may be due to the synergistic effects of the decreasing particle size from micrometer to nanometer [43] and the NC coating over the sensitive particle [11,44]. Compared to samples 3 and 4, the addition of ADN can reduce the mechanical sensitivity, apart from the reasons mentioned above; another probable reason is that ADN can lubricate the NC matrix and adsorb heat in the melting process under impact or friction.

## 4. Conclusions

In this paper, we reported on the preparation of NC/ADN-based composite fibers by the electrospinning technique. The nanofiber structures were confirmed by SEM, TEM, and other tests. Based on the TG-DSC, pressure curves, and laser ignition tests, it was found that ADN could accelerate the thermal decomposition of NC-based composite fibers. The positive oxygen balance of ADN and the intensive interactions between ADN and NC reduced the ignition delay time and promoted the burning reaction intensity of the NC-based composite fibers. The mechanical sensitivities of the NC/ADN-based composite fibers containing CL-20 or RDX were reduced. Although the composite fibers with zero OB have not been fabricated due to the feasibility problem of electrospinning, the positive influence of the OB adjustment on the burning behavior of energetic materials can be seen, and the electrospinning was an effective way to prepare high-performance NC/ADN-based composite materials.

## Figures and Tables

**Figure 1 nanomaterials-13-00717-f001:**
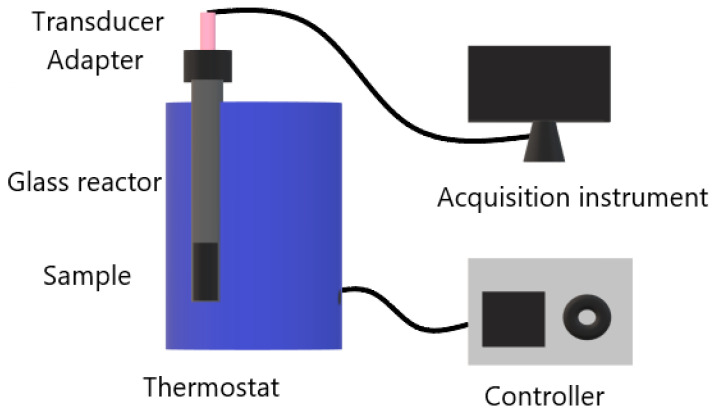
The schematic diagram of the instrument for measuring gas pressure.

**Figure 2 nanomaterials-13-00717-f002:**
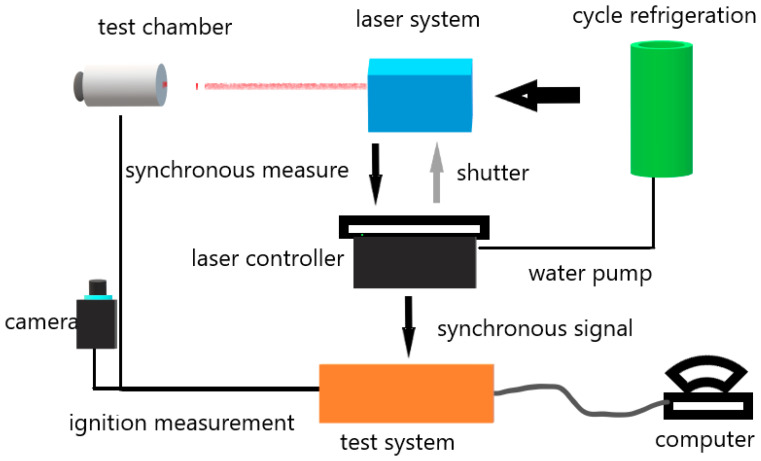
The schematic diagram of the experiment setup for combustion tests.

**Figure 3 nanomaterials-13-00717-f003:**
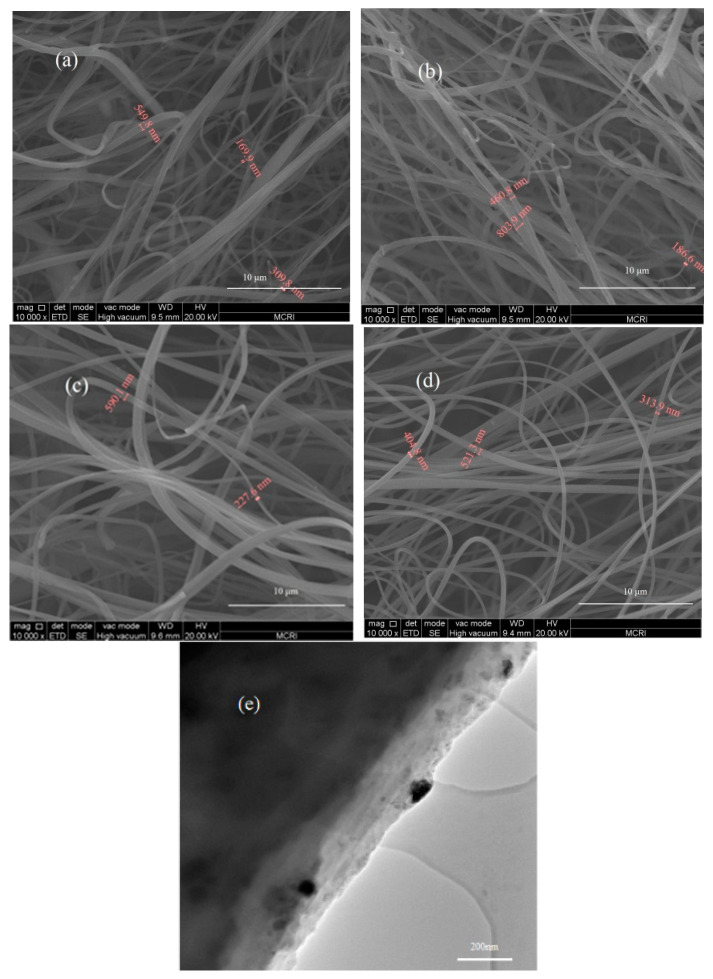
Images of electrospun NC-based composite fibers ((**a**–**d**) SEM of samples 1, 2, 3, 4; (**e**) TEM of sample 3).

**Figure 4 nanomaterials-13-00717-f004:**
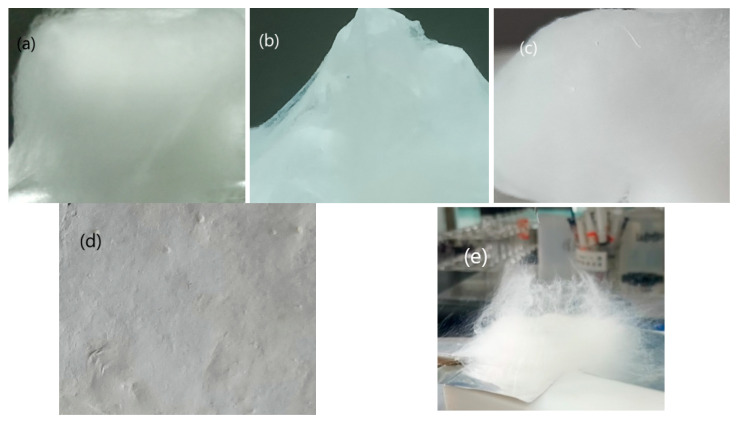
The optical photographs of electrospun NC-based composite fibers ((**a**)—sample 1, (**b**)—sample 2, (**c**)—sample 3, (**d**)—sample 4, (**e**)—the vertical growth of the fluffy NC fiber).

**Figure 5 nanomaterials-13-00717-f005:**
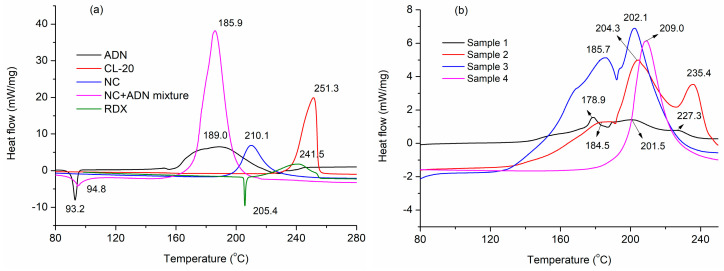
DSC curves of several single energetic materials and NC−based composite fibers. (**a**): DSC curves of explosives such as NC, ADN, NC+ADN mixture, RDX and CL-20, (**b**) DSC curves of sample 1–4.

**Figure 6 nanomaterials-13-00717-f006:**
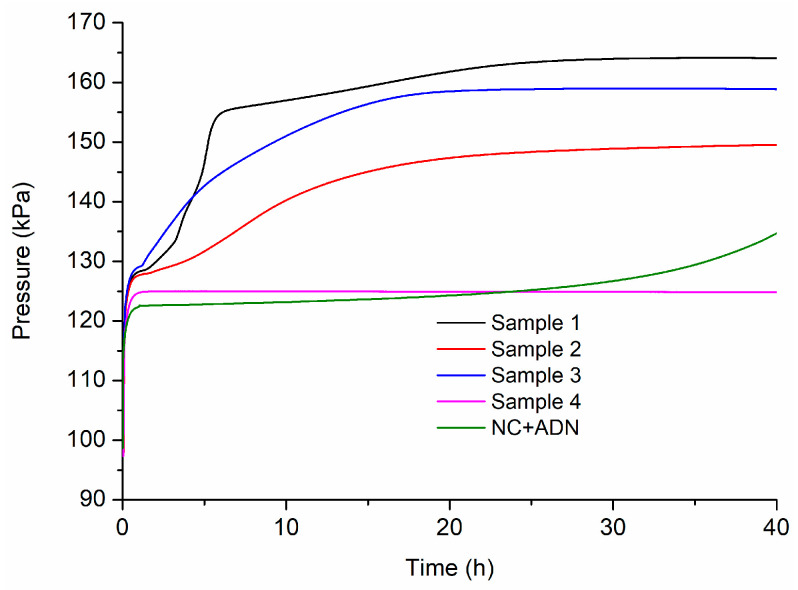
Pressure curves of NC-based composite fibers at 100 °C.

**Figure 7 nanomaterials-13-00717-f007:**
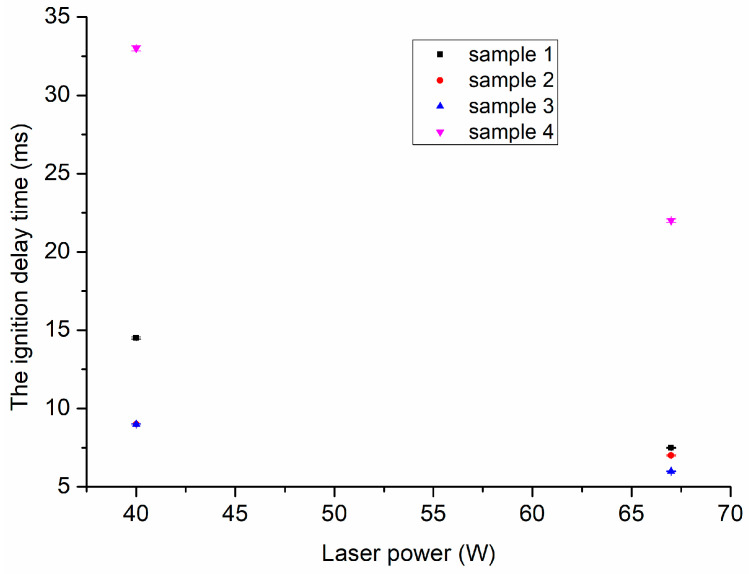
The ignition delay time of samples 1, 2, 3, and 4.

**Figure 8 nanomaterials-13-00717-f008:**
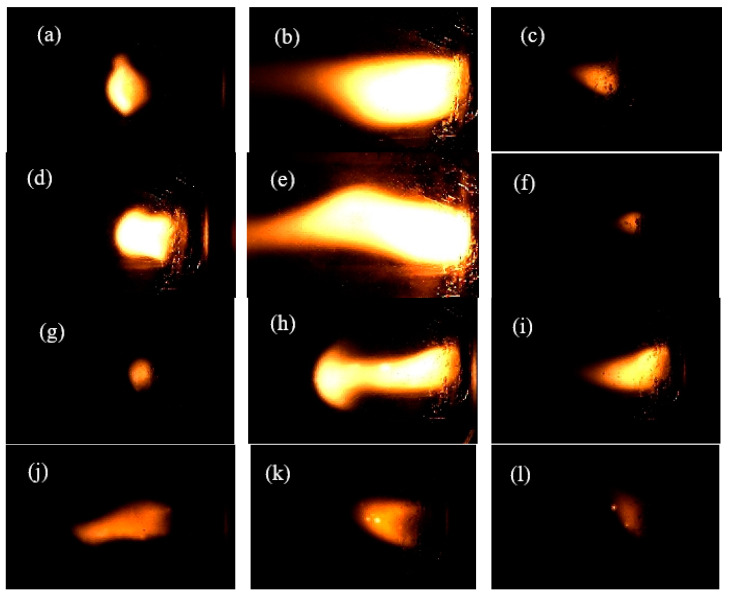
The burning snapshots of samples 1, 2, 3, and 4 under normal air pressure at laser power 40 W: (**a**) sample 1, 15 ms; (**b**) sample 1, 39 ms; (**c**) sample 1, 70 ms; (**d**) sample 2, 9 ms; (**e**) sample 2, 40 ms; (**f**) sample 2, 70 ms; (**g**) sample 3, 9 ms; (**h**) sample 3, 22 ms; (**i**) sample 3, 40 ms; (**j**) sample 4, 33 ms; (**k**) sample 4, 65 ms; (**l**) sample 4, 93 ms.

**Table 1 nanomaterials-13-00717-t001:** The components of each sample.

Sample	NC 10 wt% Solution (mg)	Ethanol (mg)	ADN (mg)	Additive–Weight (mg)	OB (%)
1	500	100	30	RDX solution–100	−16.9
2	500	100	30	CL-20 solution–100	−13.3
3	500	100	30	0	−14.5
4	500	100	0	0	−38.7

**Table 2 nanomaterials-13-00717-t002:** Characteristic data for the thermal decomposition of NC-based samples.

Sample	The First Exothermal Peak Temperature Pertaining to ADN (°C)	The Second Exothermal Peak Temperature Pertaining to NC (°C)	The Third Exothermal Peak Temperature Pertaining to Additives (°C)
4	—	209.0	—

**Table 3 nanomaterials-13-00717-t003:** The test results of mechanical sensitivity.

Sample	Impact Sensitivity (%)	Friction Sensitivity (%)
1	56	60
2	56	64
3	52	60
4	64	72
ADN	48	40
RDX	88	84
CL-20	100	100

## Data Availability

All data can be accessed from the corresponding author through email.

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
