# Peer review of "Characterization of Electrospinning Prepared Nitrocellulose (NC)-Ammonium Dinitramide (ADN)-Based Composite Fibers"

_nanomaterials, 2023, doi:10.3390/nano13040717_

Round 1
Reviewer 1 Report
Presented manuscript entitled "Characterization of electron-spinning prepared nitrocellulose (NC)-ammonium dinitramide (ADN) based composite fibers" reports the new reactivity data for NC/ADN-based energetic composites prepared by the authors. The topic is actual, some pf the results are interesting as there are non-additive and unexpected findings. Paper can be pubblished, but some revision and language polishing are required. My detailed comments from the beginning:
abstract: "Nano-scale composite energetic materials based on oxidizer and fuel have potential ad-9 vantages in energy adjustment and regulation through oxygen balance (OB) change." - unclear sentence. Every mixture of fuel and oxidized can be tuned by OB change, but here nanoscale offers new possibilities, please , revise
"and the pressure rise time moving forward." - not clear
"In addition, the impact sensitivities were re-21 duced from 100% to 56% for NC-based fibers containing 2,4,6,8,10,12- hexanitro-2,4,6,8,10,12-hex-22 anitrohexaazaiso-wurzitane (CL-20) and from 88% to 56% for NC-based fibers containing hexahy-23 dro-1,3,5-trinitro- 1,3,5-triazine (RDX), respectively." - not clear what these number refer to? what are two mixtures with CL-20 and RDX that you compare?
introduction: on advantages of nano-NC, please discuss also the important papers on the subject https://doi.org/10.3390/nano9101386; https://doi.org/10.1002/prep.202000240
on the sensitivity on nano-energetic materials, please discuss also tow important studies https://doi.org/10.1016/j.cej.2021.129804; https://doi.org/10.1002/prep.201200189
experimental "(DSC, Q200 in TA company" -> "(DSC, Q200, TA instruments"
"25 mL 96 min-1." - should be the upperscript
"diagrammatic sketch" - revise
"CO2 laser 114 (SLC110)," - please provide more details, e.g., the wavelength
"The impact sensitivity was measured on the WL-1 instrument with the hammer 124 weight 10kg and the sample mass 50mg. Totally 25 times were done for each sample, and 125 the explosion probability was calculated as the last result in the form of explosion percent-126 age." - was it the up-and-down method or the height was constant? please give the reference to the corresponding standard
results, Figure 5 - make the same scale for (a) and (b) plots, from 50C
"Taking the Tp of NC as an indicator, the thermal stability order was [sample1] < [sample 203 3] < [sample 2] < [sample 4]." - it is very strong statement. In fact, the characteristic temperature is a first, crude estimate of thermal stability, the correct comparison can be drawn only when the kinetics is considered (e.g., see the recent discussion in https://doi.org/10.1016/j.tca.2022.179384). i recommend at least to add this precaution
Figure 7 - please indicate the measurement errors on the plot
section 3.5, please give the estimated inaccuracies, is there any difference between 52 and 56%, for example
Authors contribution - please provide it in standard Credit format
Author Response
Dear Reviewer and editors,
The texts have been revised and responsed according to the comments. Thank you very much for your important suggestions, which could be for the high quality of this manuscript.
Best regards,
Weiqiang

Reviewer 2 Report
The research article on " Characterization of electron-spinning prepared nitrocellulose (NC)-ammonium dinitramide (ADN) based composite fibers" reports the synthesis of NC-ADN based composite fibers and its characteristic features. Below are some of the comments that authors should look into improving the manuscript.
1.The authors have mentioned the preparation of 4 different samples (Table 3) and there is no much change comparing the results. The authors should explain in detail about the novelty of the work?
2.The authors have fixed the ratio of ADN (ammonium dinitramide) as constant (30mg) for preparing different samples. From the results we can notice that there is no much difference in the oxygen balance with different additives. It would be better if the authors can add more results of varying the amount of ADN?
3.The DSC results shows confusion as the authors have mentioned the preparation of 4 different samples (Sample 1-4). But in Figure 5 the authors have showed the data for NC+ADN which the author has mentioned as sample 3. It would be better to mark the sample details properly and explain it in the manuscript. Likewise, authors have not shown the data of the samples 1-4 until 50°C in Figure 5?
4.The authors have mentioned in Line: 214 that “The first pressure rise process for sample 1, 2 and 3 can be ascribed to the thermal decomposition of ADN, because ADN decomposed ahead of NC as shown in DSC curves in Figure 5”. From the DSC data it can be noticed that the decomposition behaviour started only after 150°C, additionally it is well known that, if the pressure increases the decomposition rate decreases. The authors should try to explain in detail the exact reason why it shows such a behaviour?
5.In figure 7, the ignition delay time of RDX, CL20, AND-NC-30mg and AND-NC-0mg are shown. It would be better to show all the results for the rest of the samples for better understanding as there is no much difference in the displayed samples.
6.There are several grammatical errors and spelling mistakes throughout the manuscript.
Author Response

(The authors gave the same response as above.)

Reviewer 3 Report
The explosive nitrocellulose was first reported at the end of the 1890s. And as this paper demonstrates, it continues to be of interest. The authors have performed a thorough sensitivity study of it in when combined with a more modern explosive. Such studies began to be published only very recently (towards the end of the 2010s). So this study is to be welcomed. I recommend it for publication so long as the authors attend to the minor points raised below.
Line 2 (Title): Replace ‘electron-spinning’ by ‘electro-spinning’
Line 12. Replace ‘electron-spinning’ by ‘electro-spinning’
Line 17. Replace ‘imbedded’ by ‘embedded’
Line 28. Replace ‘electron-spinning’ by ‘electro-spinning’
Line 40. Replace ‘had’ by ‘have’
Line 42. Replace ‘electron-spinning’ by ‘electro-spinning’
Line 45. Replace ‘electron-spinning’ by ‘electro-spinning’
Line 60. Replace ‘electron-spinning’ by ‘electro-spinning’
Line 79. Replace ‘were’ by ‘are’
Line 81. Replace ‘electron-spinning’ by ‘electro-spinning’
Line 84. Replace ‘electron-spinning’ by ‘electro-spinning’
Line 87. Replace ‘collected’ by ‘connected’
Lines 134, 135, 140 & 144. Replace ‘electronspun’ by ‘electrospun’
Line 150. Replace ‘imbedded’ by ‘embedded’
Lines 152-153. Replace ‘electropspun’ by ‘electrospun’
Line 165. Replace ‘electron-spinning’ by ‘electro-spinning’
Line 166. Replace ‘Figure4 e’ by ‘Figure 4e’
Line 179. Replace ‘imbedded’ by ‘embedded’
Line 183. Replace ‘electron-spinning’ by ‘electro-spinning’
Line 210. Replace ‘contained’ by ‘containing’
Line 218. Replace ‘Compared’ by ‘Comparing’
Lines 225 & 226. Replace ‘electron-spinning’ by ‘electro-spinning’
Line 239. Replace ‘power laser’ by ‘laser power’
Line 293. Replace ‘ere’ by ‘were’
Line 294. Replace ‘lase’ by ‘laser’
Lines 293, 300, 301. Replace ‘electron-spinning’ by ‘electro-spinning’
Line 300. Replace ‘be’ by ‘been’
Author Response

(The authors gave the same response as above.)

Round 2
Reviewer 2 Report
I agree with all of the revisions made by authors.